# Learning Deep Structured Multi-Scale Features using Attention-Gated CRFs for Contour Prediction

**Dan Xu**[1]   **Wanli Ouyang**[2]   **Xavier Alameda-Pineda**[3]   **Elisa Ricci**[4]
**Xiaogang Wang**[5]   **Nicu Sebe**[1]

[1]The University of Trento, [2]The University of Sydney, [3]Perception Group, INRIA
[4]University of Perugia, [5]The Chinese University of Hong Kong

dan.xu@unitn.it, wanli.ouyang@sydney.edu.au, xavier.alameda-pineda@inria.fr
elisa.ricci@unipg.it, xgwang@ee.cuhk.edu.hk, niculae.sebe@unitn.it

## Abstract

Recent works have shown that exploiting multi-scale representations deeply learned via convolutional neural networks (CNN) is of tremendous importance for accurate contour detection. This paper presents a novel approach for predicting contours which advances the state of the art in two fundamental aspects, *i.e.* multi-scale feature generation and fusion. Different from previous works directly considering multi-scale feature maps obtained from the inner layers of a primary CNN architecture, we introduce a hierarchical deep model which produces more rich and complementary representations. Furthermore, to refine and robustly fuse the representations learned at different scales, the novel Attention-Gated Conditional Random Fields (AG-CRFs) are proposed. The experiments ran on two publicly available datasets (BSDS500 and NYUDv2) demonstrate the effectiveness of the latent AG-CRF model and of the overall hierarchical framework.

## 1 Introduction

Considered as one of the fundamental tasks in low-level vision, contour detection has been deeply studied in the past decades. While early works mostly focused on low-level cues (*e.g.* colors, gradients, textures) and hand-crafted features [3, 25, 22], more recent methods benefit from the representational power of deep learning models [31, 2, 38, 19, 24]. The ability to effectively exploit multi-scale feature representations is considered a crucial factor for achieving accurate predictions of contours in both traditional [29] and CNN-based [38, 19, 24] approaches. Restricting the attention on deep learning-based solutions, existing methods [38, 24] typically derive multi-scale representations by adopting standard CNN architectures and considering directly the feature maps associated to different inner layers. These maps are highly complementary: while the features from the first layers are responsible for predicting fine details, the ones from the higher layers are devoted to encode the basic structure of the objects. Traditionally, concatenation and weighted averaging are very popular strategies to combine multi-scale representations (see Fig. 1.a). While these strategies typically lead to an increased detection accuracy with respect to single-scale models, they severely simplify the complex relationship between multi-scale feature maps.

The motivational cornerstone of this study is the following research question: is it worth modeling and exploiting complex relationships between multiple scales of a deep representation for contour detection? In order to provide an answer and inspired by recent works exploiting graphical models within deep learning architectures [5, 39], we introduce *Attention-Gated Conditional Random Fields* (AG-CRFs), which allow to learn robust feature map representations at each scale by exploiting the information available from other scales. This is achieved by incorporating an attention mechanism [27] seamlessly integrated into the multi-scale learning process under the form of gates [26]. Intuitively,

the attention mechanism will further enhance the quality of the learned multi-scale representation, thus improving the overall performance of the model.

We integrated the proposed AG-CRFs into a two-level hierarchical CNN model, defining a novel Attention-guided Multi-scale Hierarchical deepNet (AMH-Net) for contour detection. The hierarchical network is able to learn richer multi-scale features than conventional CNNs, the representational power of which is further enhanced by the proposed AG-CRF model. We evaluate the effectiveness of the overall model on two publicly available datasets for the contour detection task, *i.e.* BSDS500 [1] and NYU Depth v2 [33]. The results demonstrate that our approach is able to learn rich and complementary features, thus outperforming state-of-the-art contour detection methods.

**Related work.** In the last few years several deep learning models have been proposed for detecting contours [31, 2, 41, 38, 24, 23]. Among these, some works explicitly focused on devising multi-scale CNN models in order to boost performance. For instance, the Holistically-Nested Edge Detection method [38] employed multiple side outputs derived from the inner layers of a primary CNN and combine them for the final prediction. Liu *et al.* [23] introduced a framework to learn rich deep representations by concatenating features derived from all convolutional layers of VGG16. Bertasius *et al.* [2] considered skip-layer CNNs to jointly combine feature maps from multiple layers. Maninis *et al.* [24] proposed Convolutional Oriented Boundaries (COB), where features from different layers are fused to compute oriented contours and region hierarchies. However, these works combine the multi-scale representations from different layers adopting concatenation and weighted averaging schemes while not considering the dependency between the features. Furthermore, these works do not focus on generating more rich and diverse representations at each CNN layer.

The combination of multi-scale representations has been also widely investigated for other pixel-level prediction tasks, such as semantic segmentation [43], visual saliency detection [21] and monocular depth estimation [39], and different deep architectures have been designed. For instance, to effectively aggregate the multi-scale information, Yu *et al.* [43] introduced dilated convolutions. Yang *et al.* [42] proposed DAG-CNNs where multi-scale feature outputs from different ReLU layers are combined through element-wise addition operator. However, none of these works incorporate an attention mechanism into a multi-scale structured feature learning framework.

Attention models have been successfully exploited in deep learning for various tasks such as image classification [37], speech recognition [4] and image caption generation [40]. However, to our knowledge, this work is the first to introduce an attention model for estimating contours. Furthermore, we are not aware of previous studies integrating the attention mechanism into a probabilistic (CRF) framework to control the message passing between hidden variables. We model the attention as *gates* [26], which have been used in previous deep models such as restricted Boltzman machine for unsupervised feature learning [35], LSTM for sequence learning [12, 6] and CNN for image classification [44]. However, none of these works explore the possibility of jointly learning multi-scale deep representations and an attention model within a unified probabilistic graphical model.

## 2 Attention-Gated CRFs for Deep Structured Multi-Scale Feature Learning

### 2.1 Problem Definition and Notation

Given an input image $\mathbf{I}$ and a generic front-end CNN model with parameters $\mathbf{W}_c$, we consider a set of $S$ multi-scale feature maps $\mathbf{F} = \{\mathbf{f}_s\}_{s=1}^S$. Being a generic framework, these feature maps can be the output of $S$ intermediate CNN layers or of another representation, thus $s$ is a *virtual* scale. The feature map at scale $s$, $\mathbf{f}_s$ can be interpreted as a set of feature vectors, $\mathbf{f}_s = \{\mathbf{f}_s^i\}_{i=1}^N$, where $N$ is the number of pixels. Opposite to previous works adopting simple concatenation or weighted averaging schemes [16, 38], we propose to combine the multi-scale feature maps by learning a set of latent feature maps $\mathbf{h}_s = \{\mathbf{h}_s^i\}_{i=1}^N$ with a novel *Attention-Gated CRF* model sketched in Fig.1. Intuitively, this allows a joint refinement of the features by flowing information between different scales. Moreover, since the information from one scale may or may not be relevant for the pixels at another scale, we utilise the concept of *gate*, previously introduced in the literature in the case of graphical models [36], in our CRF formulation. These gates are binary random hidden variables that permit or block the flow of information between scales at every pixel. Formally, $g_{s_e,s_r}^i \in \{0,1\}$ is the gate at pixel $i$ of scale $s_r$ (receiver) from scale $s_e$ (emitter), and we also write $\mathbf{g}_{s_e,s_r} = \{g_{s_e,s_r}^i\}_{i=1}^N$. Precisely, when $g_{s_e,s_r}^i = 1$ then the hidden variable $\mathbf{h}_{s_r}^i$ is updated taking (also) into account the

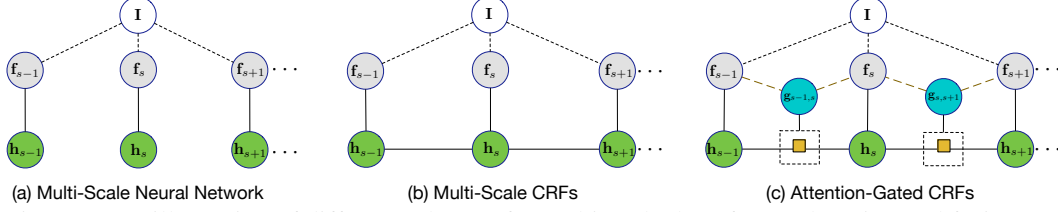

Figure 1: An illustration of different schemes for multi-scale deep feature learning and fusion. (a) the traditional approach (*e.g.* concatenation, weighted average), (b) CRF implementing multi-scale feature fusion (c) the proposed AG-CRF-based approach.

information from the $s_e$-th layer, *i.e.* $\mathbf{h}_{s_e}$. As shown in the following, the joint inference of the hidden features and the gates leads to estimating the optimal features as well as the corresponding attention model, hence the name Attention-Gated CRFs.

## 2.2 Attention-Gated CRFs

Given the observed multi-scale feature maps $\mathbf{F}$ of image $\mathbf{I}$, the objective is to estimate the hidden multi-scale representation $\mathbf{H} = \{\mathbf{h}_s\}_{s=1}^S$ and, accessorily the attention gate variables $\mathbf{G} = \{\mathbf{g}_{s_e,s_r}\}_{s_e,s_r=1}^S$. To do that, we formalize the problem within a conditional random field framework and write the Gibbs distribution as $P(\mathbf{H}, \mathbf{G}|\mathbf{I}, \Theta) = \exp\left(-E(\mathbf{H}, \mathbf{G}, \mathbf{I}, \Theta)\right)/Z(\mathbf{I}, \Theta)$, where $\Theta$ is the set of parameters and $E$ is the energy function. As usual, we exploit both unary and binary potentials to couple the hidden variables between them and to the observations. Importantly, the proposed binary potential is gated, and thus only active when the gate is open. More formally the general form[1] of the energy function writes:

$$E(\mathbf{H}, \mathbf{G}, \mathbf{I}, \Theta) = \underbrace{\sum_s \sum_i \phi_h(\mathbf{h}_s^i, \mathbf{f}_s^i)}_{\text{Unary potential}} + \underbrace{\sum_{s_e,s_r} \sum_{i,j} g_{s_e,s_r}^i \psi_h(\mathbf{h}_{s_r}^i, \mathbf{h}_{s_e}^j)}_{\text{Gated pairwise potential}}. \tag{1}$$

The first term of the energy function is a classical unary term that relates the hidden features to the observed multi-scale CNN representations. The second term synthesizes the theoretical contribution of the present study because it conditions the effect of the pair-wise potential $\psi_h(\mathbf{h}_{s_e}^i, \mathbf{h}_{s_r}^j)$ upon the gate hidden variable $g_{s_e,s_r}^i$. Fig. 1c depicts the model formulated in Equ.(1). If we remove the attention gate variables, it becomes a general multi-scale CRFs as shown in Fig. 1b.

Given that formulation, and as it is typically the case in conditional random fields, we exploit the mean-field approximation in order to derive a tractable inference procedure. Under this generic form, the mean-field inference procedure writes:

$$q(\mathbf{h}_s^i) \propto \exp\left(\phi_h(\mathbf{h}_s^i, \mathbf{f}_s^i) + \sum_{s' \neq s} \sum_j \mathbb{E}_{q(g_{s',s}^i)}\{g_{s',s}^i\} \mathbb{E}_{q(\mathbf{h}_{s'}^j)}\{\psi_h(\mathbf{h}_s^i, \mathbf{h}_{s'}^j)\}\right), \tag{2}$$

$$q(g_{s',s}^i) \propto \exp\left(g_{s',s}^i \mathbb{E}_{q(\mathbf{h}_s^i)}\left\{\sum_j \mathbb{E}_{q(\mathbf{h}_{s'}^j)}\left\{\psi_h(\mathbf{h}_s^i, \mathbf{h}_{s'}^j)\right\}\right\}\right), \tag{3}$$

where $\mathbb{E}_q$ stands for the expectation with respect to the distribution $q$.

Before deriving these formulae for our precise choice of potentials, we remark that, since the gate is a binary variable, the expectation of its value is the same as $q(g_{s',s}^i = 1)$. By defining: $\mathcal{M}_{s',s}^i = \mathbb{E}_{q(\mathbf{h}_s^i)}\left\{\sum_j \mathbb{E}_{q(\mathbf{h}_{s'}^j)}\left\{\psi_h(\mathbf{h}_s^i, \mathbf{h}_{s'}^j)\right\}\right\}$, the expected value of the gate writes:

$$\alpha_{s,s'}^i = \mathbb{E}_{q(g_{s',s}^i)}\{g_{s',s}^i\} = \frac{q(g_{s',s}^i = 1)}{q(g_{s',s}^i = 0) + q(g_{s',s}^i = 1)} = \sigma\left(-\mathcal{M}_{s',s}^i\right), \tag{4}$$

where $\sigma()$ denotes the sigmoid function. This finding is specially relevant in the framework of CNN since many of the attention models are typically obtained after applying the sigmoid function to the

features derived from a feed-forward network. Importantly, since the quantity $\mathcal{M}^i_{s',s}$ depends on the expected values of the hidden features $\mathbf{h}^i_s$, the AG-CRF framework extends the unidirectional connection from the features to the attention model, to a bidirectional connection in which the expected value of the gate allows to refine the distribution of the hidden features as well.

## 2.3 AG-CRF Inference

In order to construct an operative model we need to define the unary and gated potentials $\phi_h$ and $\psi_h$. In our case, the unary potential corresponds to an isotropic Gaussian:

$$\phi_h(\mathbf{h}^i_s, \mathbf{f}^i_s) = -\frac{a^i_s}{2}\|\mathbf{h}^i_s - \mathbf{f}^i_s\|^2, \tag{5}$$

where $a^i_s > 0$ is a weighting factor.

The gated binary potential is specifically designed for a two-fold objective. On the one hand, we would like to learn and further exploit the relationships between hidden vectors at the same, as well as at different scales. On the other hand, we would like to exploit previous knowledge on attention models and include linear terms in the potential. Indeed, this would implicitly shape the gate variable to include a linear operator on the features. Therefore, we chose a bilinear potential:

$$\psi_h(\mathbf{h}^i_s, \mathbf{h}^j_{s'}) = \tilde{\mathbf{h}}^i_s \mathbf{K}^{i,j}_{s,s'} \tilde{\mathbf{h}}^j_{s'}, \tag{6}$$

where $\tilde{\mathbf{h}}^i_s = (\mathbf{h}^{i\top}_s, 1)^\top$ and $\mathbf{K}^{i,j}_{s,s'} \in \mathbb{R}^{(C_s+1)\times(C_{s'}+1)}$ being $C_s$ the size, i.e. the number of channels, of the representation at scale $s$. If we write this matrix as $\mathbf{K}^{i,j}_{s,s'} = (\mathbf{L}^{i,j}_{s,s'}, \mathbf{l}^{i,j}_{s,s'}; \mathbf{l}^{j,i\top}_{s',s}, 1)$, then $\mathbf{L}^{i,j}_{s,s'}$ exploits the relationships between hidden variables, while $\mathbf{l}^{i,j}_{s,s'}$ and $\mathbf{l}^{j,i}_{s',s}$ implement the classically used linear relationships of the attention models. In order words, $\psi_h$ models the pair-wise relationships between features with the upper-left block of the matrix. Furthemore, $\psi_h$ takes into account the linear relationships by completing the hidden vectors with the unity. In all, the energy function writes:

$$E(\mathbf{H}, \mathbf{G}, \mathbf{I}, \Theta) = -\sum_s \sum_i \frac{a^i_s}{2}\|\mathbf{h}^i_s - \mathbf{f}^i_s\|^2 + \sum_{s_e,s_r}\sum_{i,j} g^i_{s_e,s_r} \tilde{\mathbf{h}}^i_{s_r} \mathbf{K}^{i,j}_{s_r,s_e} \tilde{\mathbf{h}}^j_{s_e}. \tag{7}$$

Under these potentials, we can consequently update the mean-field inference equations to:

$$q(\mathbf{h}^i_s) \propto \exp\left(-\frac{a^i_s}{2}(\|\mathbf{h}^i_s\| - 2\mathbf{h}^{i\top}_s\mathbf{f}^i_s) + \sum_{s'\neq s} \alpha^i_{s,s'}\mathbf{h}^{i\top}_s \sum_j (\mathbf{L}^{i,j}_{s,s'}\bar{\mathbf{h}}^j_{s'} + \mathbf{l}^{i,j}_{s,s'})\right), \tag{8}$$

where $\bar{\mathbf{h}}^j_{s'}$ is the expected a posteriori value of $\mathbf{h}^j_{s'}$.

The previous expression implies that the a posteriori distribution for $\mathbf{h}^i_s$ is a Gaussian. The mean vector of the Gaussian and the function $\mathcal{M}$ write:

$$\bar{\mathbf{h}}^i_s = \frac{1}{a^i_s}\left(a^i_s\mathbf{f}^i_s + \sum_{s'\neq s}\alpha^i_{s,s'}\sum_j(\mathbf{L}^{i,j}_{s,s'}\bar{\mathbf{h}}^j_{s'}+\mathbf{l}^{i,j}_{s,s'})\right) \quad \mathcal{M}^i_{s',s} = \sum_j\left(\bar{\mathbf{h}}^i_s\mathbf{L}^{i,j}_{s,s'}\bar{\mathbf{h}}^j_{s'} + \bar{\mathbf{h}}^{i\top}_s\mathbf{l}^{i,j}_{s,s'} + \bar{\mathbf{h}}^{j\top}_{s'}\mathbf{l}^{j,i}_{s',s}\right)$$

which concludes the inference procedure. Furthermore, the proposed framework can be simplified to obtain the traditional attention models. In most of the previous studies, the attention variables are computed directly from the multi-scale features instead of computing them from the hidden variables. Indeed, since many of these studies do not propose a probabilistic formulation, there are no hidden variables and the attention is computed sequentially through the scales. We can emulate the same behavior within the AG-CRF framework by modifying the gated potential as follows:

$$\tilde{\psi}_h(\mathbf{h}^i_s, \mathbf{h}^j_{s'}, \mathbf{f}^i_s, \mathbf{f}^j_{s'}) = \mathbf{h}^i_s\mathbf{L}^{i,j}_{s,s'}\mathbf{h}^j_{s'} + \mathbf{f}^{i\top}_s\mathbf{l}^{i,j}_{s,s'} + \mathbf{f}^{j\top}_{s'}\mathbf{l}^{j,i}_{s',s}. \tag{9}$$

This means that we keep the pair-wise relationships between hidden variables (as in any CRF) and let the attention model be generated by a linear combination of the observed features from the CNN, as it is traditionally done. The changes in the inference procedure are straightforward and reported in the supplementary material due to space constraints. We refer to this model as partially-latent AG-CRFs (PLAG-CRFs), whereas the more general one is denoted as fully-latent AG-CRFs (FLAG-CRFs).

## 2.4 Implementation with neural network for joint learning

In order to infer the hidden variables and learn the parameters of the AG-CRFs together with those of the front-end CNN, we implement the AG-CRFs updates in neural network with several steps:

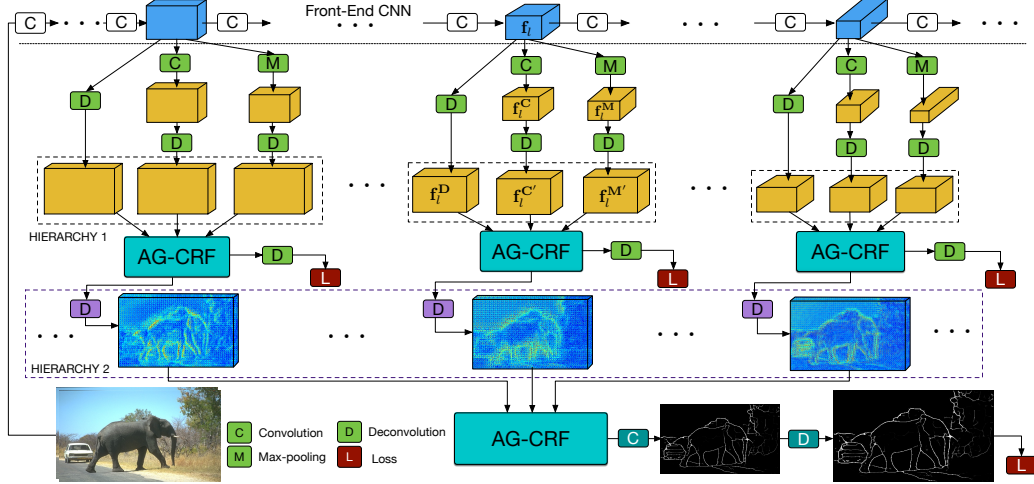

Figure 2: An overview of the proposed AMH-Net for contour detection.

(i) message passing from the $s_e$-th scale to the current $s_r$-th scale is performed with $\mathbf{h}_{s_e \to s_r} \leftarrow \mathbf{L}_{s_e \to s_r} \otimes \mathbf{h}_{s_e}$, where $\otimes$ denotes the convolutional operation and $\mathbf{L}_{s_e \to s_r}$ denotes the corresponding convolution kernel, (ii) attention map estimation $q(\mathbf{g}_{s_e, s_r} = \mathbf{1}) \leftarrow \sigma(\mathbf{h}_{s_r} \odot (\mathbf{L}_{s_e \to s_r} \otimes \mathbf{h}_{s_e}) + \mathbf{l}_{s_e \to s_r} \otimes \mathbf{h}_{s_e} + \mathbf{l}_{s_r \to s_e} \otimes \mathbf{h}_{s_r})$, where $\mathbf{L}_{s_e \to s_r}$, $\mathbf{l}_{s_e \to s_r}$ and $\mathbf{l}_{s_r \to s_e}$ are convolution kernels and $\odot$ represents element-wise product operation, and (iii) attention-gated message passing from other scales and adding unary term: $\mathbf{h}_{s_r} = \mathbf{f}_{s_r} \oplus a_{s_r} \sum_{s_e \neq s_r} (q(\mathbf{g}_{s_e, s_r} = \mathbf{1}) \odot \mathbf{h}_{s_e \to s_r})$, where $a_{s_r}$ encodes the effect of the $a_{s_r}^i$ for weighting the message and can be implemented as a $1 \times 1$ convolution. The symbol $\oplus$ denotes element-wise addition. In order to simplify the overall inference procedure, and because the magnitude of the linear term of $\psi_h$ is in practice negligible compared to the quadratic term, we discard the message associated to the linear term. When the inference is complete, the final estimate is obtained by convolving all the scales.

## 3   Exploiting AG-CRFs with a Multi-scale Hierarchical Network

**AMH-Net Architecture.** The proposed Attention-guided Multi-scale Hierarchical Network (AMH-Net), as sketched in Figure 2, consists of a multi-scale hierarchical network (MH-Net) together with the AG-CRF model described above. The MH-Net is constructed from a front-end CNN architecture such as the widely used AlexNet [20], VGG [34] and ResNet [17]. One prominent feature of MH-Net is its ability to generate richer multi-scale representations. In order to do that, we perform distinct non-linear mappings (deconvolution $\mathbf{D}$, convolution $\mathbf{C}$ and max-pooling $\mathbf{M}$) upon $\mathbf{f}_l$, the CNN feature representation from an intermediate layer $l$ of the front-end CNN. This leads to a three-way representation: $\mathbf{f}_l^{\mathbf{D}}$, $\mathbf{f}_l^{\mathbf{C}}$ and $\mathbf{f}_l^{\mathbf{M}}$. Remarkably, while $\mathbf{D}$ upsamples the feature map, $\mathbf{C}$ maintains its original size and $\mathbf{M}$ reduces it, and different kernel size is utilized for them to have different receptive fields, then naturally obtaining complementary inter- and multi-scale representations. The $\mathbf{f}_l^{\mathbf{C}}$ and $\mathbf{f}_l^{\mathbf{M}}$ are further aligned to the dimensions of the feature map $\mathbf{f}_l^{\mathbf{D}}$ by the deconvolutional operation. The hierarchy is implemented in two levels. The first level uses an AG-CRF model to fuse the three representations of each layer $l$, thus refining the CNN features within the same scale. The second level of the hierarchy uses an AG-CRF model to fuse the information coming from multiple CNN layers. The proposed hierarchical multi-scale structure is general purpose and able to involve an arbitrary number of layers and of diverse intra-layer representations.

**End-to-End Network Optimization.** The parameters of the model consist of the front-end CNN parameters, $\mathbf{W}_c$, the parameters to produce the richer decomposition from each layer $l$, $\mathbf{W}_l$, the parameters of the AG-CRFs of the first level of the hierarchy, $\{\mathbf{W}_l^{\mathrm{I}}\}_{l=1}^L$, and the parameters of the AG-CRFs of the second level of the hierarchy, $\mathbf{W}^{\mathrm{II}}$. $L$ is the number of intermediate layers used from the front-end CNN. In order to jointly optimize all these parameters we adopt deep supervision [38] and we add an optimization loss associated to each AG-CRF module. In addition, since the contour detection problem is highly unbalanced, *i.e.* contour pixels are significantly less than non-contour pixels, we employ the modified cross-entropy loss function of [38]. Given a training data

set $\mathcal{D} = \{(\mathbf{I}_p, \mathbf{E}_p)\}_{p=1}^P$ consisting of $P$ RGB-contour groundtruth pairs, the loss function $\ell$ writes:

$$\ell(\mathbf{W}) = \sum_p \beta \sum_{e_p^k \in \mathbf{E}_p^+} \log \mathrm{P}\big(e_p^k = 1 | \mathbf{I}_p; \mathbf{W}\big) + \big(1 - \beta\big) \sum_{e_p^k \in \mathbf{E}_p^-} \log \mathrm{P}\big(e_p^k = 0 | \mathbf{I}_p; \mathbf{W}\big), \qquad (10)$$

where $\beta = |\mathbf{E}_p^+|/(|\mathbf{E}_p^+| + |\mathbf{E}_p^-|)$, $\mathbf{E}_p^+$ is the set of contour pixels of image $p$ and $\mathbf{W}$ is the set of all parameters. The optimization is performed via the back-propagation algorithm with standard stochastic gradient descent.

**AMH-Net for contour detection.** After training of the whole AMH-Net, the optimized network parameters $\mathbf{W}$ are used for the contour detection task. Given a new test image $\mathbf{I}$, the $L + 1$ classifiers produce a set of contour prediction maps $\{\hat{\mathbf{E}}_l\}_{l=1}^{L+1} = \mathrm{AMH\text{-}Net}(\mathbf{I}; \mathbf{W})$. The $\hat{\mathbf{E}}_l$ are obtained from the AG-CRFs with elementary operations as detailed in the supplementary material. We inspire from [38] to fuse the multiple scale predictions thus obtaining an average prediction $\hat{\mathbf{E}} = \sum_l \hat{\mathbf{E}}_l/(L + 1)$.

# 4 Experiments

## 4.1 Experimental Setup

**Datasets.** To evaluate the proposed approach we employ two different benchmarks: the BSDS500 and the NYUDv2 datasets. The BSDS500 dataset is an extended dataset based on BSDS300 [1]. It consists of 200 training, 100 validation and 200 testing images. The groundtruth pixel-level labels for each sample are derived considering multiple annotators. Following [38, 41], we use all the training and validation images for learning the proposed model and perform data augmentation as described in [38]. The NYUDv2 [33] contains 1449 RGB-D images and it is split into three subsets, comprising 381 training, 414 validation and 654 testing images. Following [38] in our experiments we employ images at full resolution (*i.e.* $560 \times 425$ pixels) both in the training and in the testing phases.

**Evaluation Metrics.** During the test phase standard non-maximum suppression (NMS) [9] is first applied to produce thinned contour maps. We then evaluate the detection performance of our approach according to different metrics, including the F-measure at Optimal Dataset Scale (ODS) and Optimal Image Scale (OIS) and the Average Precision (AP). The maximum tolerance allowed for correct matches of edge predictions to the ground truth is set to 0.0075 for the BSDS500 dataset, and to .011 for the NYUDv2 dataset as in previous works [9, 14, 38].

**Implementation Details.** The proposed AMH-Net is implemented under the deep learning framework *Caffe* [18]. The implementation code is available on Github[2]. The training and testing phase are carried out on an Nvidia Titan X GPU with 12GB memory. The ResNet50 network pretrained on ImageNet [8] is used to initialize the front-end CNN of AMH-Net. Due to memory constraints, our implementation only considers three scales, *i.e.* we generate multi-scale features from three different layers of the front-end CNN (*i.e. res3d*, *res4f*, *res5c*). In our CRF model we consider dependencies between all scales. Within the AG-CRFs, the kernel size for all convolutional operations is set to $3 \times 3$ with stride 1 and padding 1. To simplify the model optimization, the parameters $a_{s_r}^i$ are set as 0.1 for all scales during training. We choose this value as it corresponds to the best performance after cross-validation in the range $[0, 1]$. The initial learning rate is set to 1e-7 in all our experiments, and decreases 10 times after every 10k iterations. The total number of iterations for BSDS500 and NYUD v2 is 40k and 30k, respectively. The momentum and weight decay parameters are set to 0.9 and 0.0002, as in [38]. As the training images have different resolution, we need to set the batch size to 1, and for the sake of smooth convergence we updated the parameters only every 10 iterations.

## 4.2 Experimental Results

In this section, we present the results of our evaluation, comparing our approach with several state of the art methods. We further conduct an in-depth analysis of our method, to show the impact of different components on the detection performance.

**Comparison with state of the art methods.** We first consider the BSDS500 dataset and compare the performance of our approach with several traditional contour detection methods, including Felz-Hut [11], MeanShift [7], Normalized Cuts [32], ISCRA [30], gPb-ucm [1], SketchTokens [22],

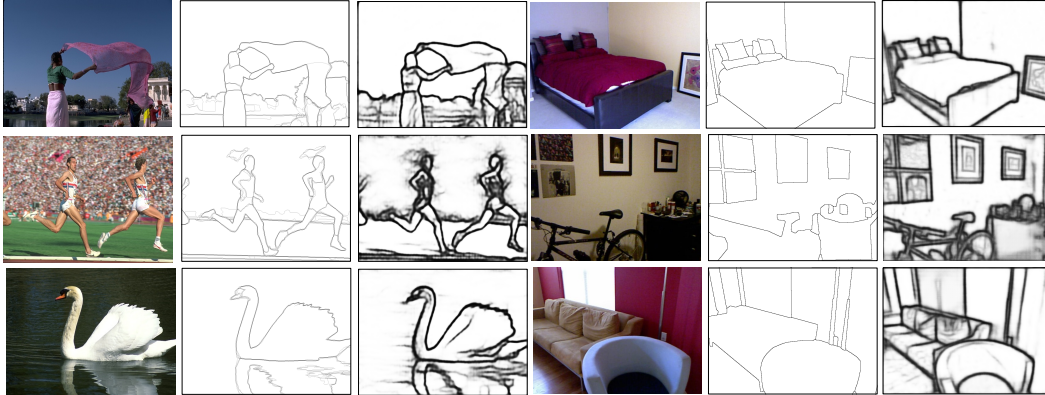

Figure 3: Qualitative results on the BSDS500 (left) and the NYUDv2 (right) test samples. The 2nd (4th) and 3rd (6th) columns are the ground-truth and estimated contour maps respectively.

Table 1: BSDS500 dataset: quantitative results.

| Method | ODS | OIS | AP |
|---|---|---|---|
| Human | .800 | .800 | - |
| Felz-Hutt[11] | .610 | .640 | .560 |
| Mean Shift[7] | .640 | .680 | .560 |
| Normalized Cuts[32] | .641 | .674 | .447 |
| ISCRA[30] | .724 | .752 | .783 |
| gPb-ucm[1] | .726 | .760 | .727 |
| Sketch Tokens[22] | .727 | .746 | .780 |
| MCG[28] | .747 | .779 | .759 |
| DeepEdge[2] | .753 | .772 | .807 |
| DeepContour[31] | .756 | .773 | .797 |
| LEP[46] | .757 | .793 | .828 |
| HED[38] | .788 | .808 | .840 |
| CEDN[41] | .788 | .804 | .834 |
| COB [24] | .793 | .820 | .859 |
| RCF [23] (not comp.) | .811 | .830 | – |
| AMH-Net (fusion) | **.798** | **.829** | **.869** |

Table 2: NYUDv2 dataset: quantitative results.

| Method | ODS | OIS | AP |
|---|---|---|---|
| gPb-ucm [1] | .632 | .661 | .562 |
| OEF [15] | .651 | .667 | – |
| Silberman *et al.* [33] | .658 | .661 | – |
| SemiContour [45] | .680 | .700 | .690 |
| SE [10] | .685 | .699 | .679 |
| gPb+NG [13] | .687 | .716 | .629 |
| SE+NG+ [14] | .710 | .723 | .738 |
| HED (RGB) [38] | .720 | .734 | .734 |
| HED (HHA) [38] | .682 | .695 | .702 |
| HED (RGB + HHA) [38] | .746 | .761 | .786 |
| RCF (RGB) + HHA) [23] | .757 | .771 | – |
| AMH-Net (RGB) | .744 | .758 | .765 |
| AMH-Net (HHA) | .716 | .729 | .734 |
| AMH-Net (RGB+HHA) | **.771** | **.786** | **.802** |

MCG [28], LEP [46], and more recent CNN-based methods, including DeepEdge [2], DeepContour [31], HED [38], CEDN [41], COB [24]. We also report results of the RCF method [23], although they are not comparable because in [23] an extra dataset (Pascal Context) was used during RCF training to improve the results on BSDS500. In this series of experiments we consider AMH-Net with FLAG-CRFs. The results of this comparison are shown in Table 1 and Fig. 4a. AMH-Net obtains an F-measure (ODS) of 0.798, thus outperforms all previous methods. The improvement over the second and third best approaches, *i.e.* COB and HED, is 0.5% and 1.0%, respectively, which is not trivial to achieve on this challenging dataset. Furthermore, when considering the OIS and AP metrics, our approach is also better, with a clear performance gap.

To perform experiments on NYUDv2, following previous works [38] we consider three different types of input representations, *i.e.* RGB, HHA [14] and RGB-HHA data. The results corresponding to the use of both RGB and HHA data (*i.e.* RGB+HHA) are obtained by performing a weighted average of the estimates obtained from two AMH-Net models trained separately on RGB and HHA representations. As baselines we consider gPb-ucm [1], OEF [15], the method in [33], SemiContour [45], SE [10], gPb+NG [13], SE+NG+ [14], HED [38] and RCF [23]. In this case the results are comparable to the RCF [23] since the experimental protocol is exactly the same. All of them are reported in Table 2 and Fig. 4b. Again, our approach outperforms all previous methods. In particular, the increased performance with respect to HED [38] and RCF [23] confirms the benefit of the proposed multi-scale feature learning and fusion scheme. Examples of qualitative results on the BSDS500 and the NYUDv2 datasets are shown in Fig. 3.

**Ablation Study.** To further demonstrate the effectiveness of the proposed model and analyze the impact of the different components of AMH-Net on the countour detection task, we conduct an

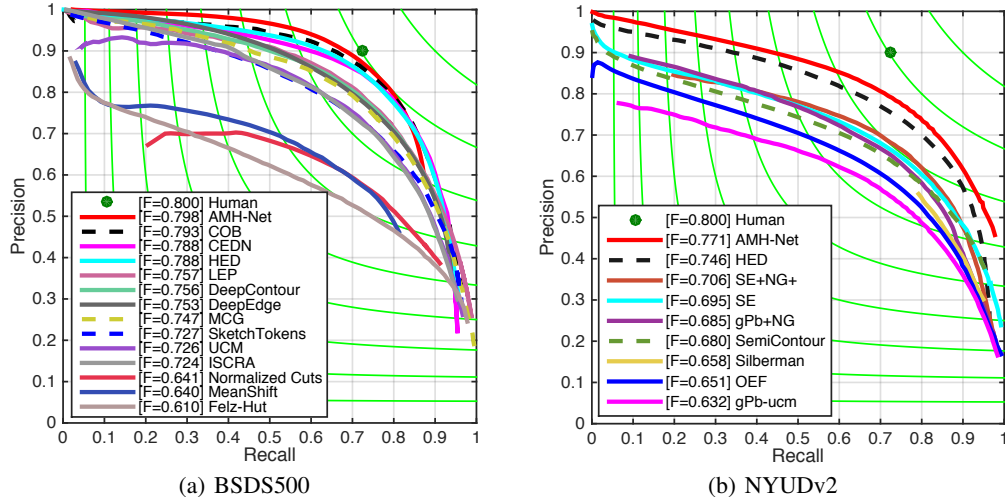

|                  | (a) BSDS500 | (b) NYUDv2 |
| ---------------- | ----------- | ---------- |

Figure 4: Precision-Recall Curves on the BSDS500 and NYUDv2 test sets.

ablation study considering the NYUDv2 dataset (RGB data). We tested the following models: (i) AMH-Net (baseline), which removes the first-level hierarchy and directly concatenates the feature maps for prediction, (ii) AMH-Net (w/o AG-CRFs), which employs the proposed multi-scale hierarchical structure but discards the AG-CRFs, (iii) AMH-Net (w/ CRFs), obtained by replacing our AG-CRFs with a multi-scale CRF model without attention gating, (iv) AMH-Net (w/o deep supervision) obtained removing intermediate loss functions in AMH-Net and (v) AMH-Net with the proposed two versions of the AG-CRFs model, *i.e.* PLAG-CRFs and FLAG-CRFs. The results of our comparison are shown in Table 3, where we also consider as reference traditional multi-scale deep learning models employing multi-scale representations, *i.e.* Hypercolumn [16] and HED [38].

These results clearly show the advantages of our contributions. The ODS F-measure of AMH-Net (w/o AG-CRFs) is 1.1% higher than AMH-Net (baseline), clearly demonstrating the effectiveness of the proposed hierarchical network and confirming our intuition that exploiting more richer and diverse multi-scale representations is beneficial. Table 3 also shows that our AG-CRFs plays a fundamental role for accurate detection, as AMH-Net (w/ FLAG-CRFs) leads to an improvement of 1.9% over AMH-Net (w/o

Table 3: Performance analysis on NYUDv2 RGB data.

| Method | ODS | OIS | AP |
| --- | --- | --- | --- |
| Hypercolumn [16] | .718 | .729 | .731 |
| HED [38] | .720 | .734 | .734 |
| AMH-Net (baseline) | .711 | .720 | .724 |
| AMH-Net (w/o AG-CRFs) | .722 | .732 | .739 |
| AMH-Net (w/ CRFs) | .732 | .742 | .750 |
| AMH-Net (w/o deep supervision) | .725 | .738 | .747 |
| AMH-Net (w/ PLAG-CRFs) | .737 | .749 | .746 |
| AMH-Net (w/ FLAG-CRFs) | **.744** | **.758** | **.765** |

AG-CRFs) in terms of OSD. Finally, AMH-Net (w/ FLAG-CRFs) is 1.2% and 1.5% better than AMH-Net (w/ CRFs) in ODS and AP metrics respectively, confirming the effectiveness of embedding an attention mechanism in the multi-scale CRF model. AMH-Net (w/o deep supervision) decreases the overall performance of our method by 1.9% in ODS, showing the crucial importance of deep supervision for better optimization of the whole AMH-Net. Comparing the performance of the proposed two versions of the AG-CRF model, *i.e.* PLAG-CRFs and FLAG-CRFs, we can see that AMH-Net (FLAG-CRFs) slightly outperforms AMH-Net (PLAG-CRFs) in both ODS and OIS, while bringing a significant improvement (around 2%) in AP. Finally, considering HED [38] and Hypercolumn [16], it is clear that our AMH-Net (FLAG-CRFs) is significantly better than these methods. Importantly, our approach utilizes only three scales while for HED [38] and Hypercolumn [16] we consider five scales. We believe that our accuracy could be further boosted by involving more scales.

## 5    Conclusions

We presented a novel multi-scale convolutional neural network for contour detection. The proposed model introduces two main components, *i.e.* a hierarchical architecture for generating more rich and complementary multi-scale feature representations, and an Attention-Gated CRF model for robust feature refinement and fusion. The effectiveness of our approach is demonstrated through extensive experiments on two public available datasets and state of the art detection performance is

achieved. The proposed approach addresses a general problem, *i.e.* how to generate rich multi-scale representations and optimally fuse them. Therefore, we believe it may be also useful for other pixel-level tasks.

## Footnotes

[1]One could certainly include a unary potential for the gate variables as well. However this would imply that there is a way to set/learn the a priori distribution of opening/closing a gate. In practice we did not observe any notable difference between using or skipping the unary potential on $g$.

[2]https://github.com/danxuhk/AttentionGatedMulti-ScaleFeatureLearning

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
