[Reviews · NeurIPS 2017]

Reviewer 1



This paper presents an interesting approach for structured prediction by introducing gated attention model in learning and fusing the multi-scale features for image labeling. The method was specifically illustrated on edge/boundary detection and promising results have been demonstrated. Using both gating function and attention models have been very popular lately but this paper tackles the multi-scale learning and fusion problem from a new angle. The proposed algorithm is well motivated and a reasonable formulation within fully convolutional neural networks. Promising results on various edge detection benchmarks have been shown. However, it will be more convincing if the generic task of image labeling such as PASCAL and COCO can be shown. This is an interesting paper with a good idea and promising results. It will be more convincing to see how it is applied to the generic image labeling task.

Reviewer 2



The main contribution of this paper is adding gates to a probabilistic graphical model to control message passing. Application wise, the author claims they're the first paper adding attention to contour detection. In terms of the contribution, as far as I know, there're already previous works on adding gates, or attention into probabilistic graphical model in a variety of ways. I just listed a few here: [1] How to generate realistic images using gated MRF’s [2] Modeling Natural Images Using Gated MRFs [3] Semantic Object Parsing with Graph LSTM [4] Structure Inference Machines: Recurrent Neural Networks for Analyzing Relations in Group Activity Recognition [5] Gated Graph Sequence Neural Networks [6] Interpretable Structure-Evolving LSTM The author did a good rebuttal. I agree that this paper provided a new type of feature CRF with mathematically grounded attention. Although the model has quite a overlap with previous works, it still provides some new ideas. The experimental results also show corresponding improvement. Authors should express the model and figure in a more clear way though (e.g. the chain graph). The hidden variable "h", label "y" should all be in the formulation.

Reviewer 3



This paper proposes a gating mechanism to combine features from different levels in a CNN for the task of contour detection. The paper builds upon recent advances in using graphical models with CNN architectures [5,39] and augments these with attention. This results in large improvements in performance at the task of contour detection, achieving new state-of-the-art results. The paper also presents an ablation study where they analyze the impact of different parts of their architecture. Cons: 1. Unclear relationship to past works which use CRFs with CNNs [5,39] and other works such as [A,B] which express CRF inference as CNNs. The paper says it is inspired from [5,39] but does not describe the points of difference from [5, 39]. Is only attention the difference and thus the novelty? Does the derivation in Section 2 follow directly from [5,39] (except taking attention into account) or it requires additional tricks and techniques beyond [5,39]? How does the derivation relate to the concepts presented in [A]? Not only will these relationships make it easier to judge novelty, it will also allow readers to contextualize the work better. 2. Intuitively, I dont understand why we need attention to combine the information from different scales. The paper is fairly vague about the motivation for this. It would help if the rebuttal could provide some examples or more intuition about what a gating mechanism can provide that a non-gating mechanism can't provide. 3. The paper presents very thorough experimental results for contour detection, but I will encourage the authors to also study at the very least one other task (say semantic segmentation). This will help evaluate how generalizable the technique is. This will also increase the impact of the paper in the community. 4. The paper is somewhat hard to read. I believe there are some undefined variables, I haven't been able to figure out what f_l^{D,C,M} in figure 2 are. 5. There are minor typos in Table 2 (performance for [14] is misreported, ODS is either 70.25 or 71.03 in [14] but the current paper reports it as 64.1), and these numbers are not consistent with the ones in Figure 4. I would recommend that the authors cross-check all numbers in the draft. [A] Efficient Inference in Fully Connected CRFs with Gaussian Edge Potentials Philipp Krähenbühl and Vladlen Koltun NIPS 2011 [B] Conditional Random Fields as Recurrent Neural Networks. ICCV 2015